# Nonlinearity-Induced Asymmetric Synchronization Region in Micromechanical Oscillators

**DOI:** 10.3390/mi15020238

**Published:** 2024-02-04

**Authors:** Zhonghua Liu, Bingchan Qin, Zhan Shi, Xuefeng Wang, Qiangfeng Lv, Xueyong Wei, Ronghua Huan

**Affiliations:** 1Department of Civil Engineering, Xiamen University, Xiamen 361005, China; liuzh@xmu.edu.cn (Z.L.); 25320211152331@stu.xmu.edu.cn (B.Q.); 2Department of Mechanics, Key Laboratory of Soft Machines and Smart Devices of Zhejiang Province, Zhejiang University, Hangzhou 310027, China; qflv@zju.edu.cn; 3Department of Engineering Mechanics, MIIT Key Laboratory of Dynamics and Control of Complex Systems, Northwestern Polytechnical University, Xi’an 710072, China; xfwang@nwpu.edu.cn; 4Huanjiang Laboratory, Zhuji 311800, China; 5State Key Laboratory for Manufacturing Systems Engineering, Xi’an Jiaotong University, Xi’an 710049, China; seanwei@mail.xjtu.edu.cn

**Keywords:** MEMS, synchronization, asymmetry synchronization region, nonlinear dynamics

## Abstract

Synchronization in microstructures is a widely explored domain due to its diverse dynamic traits and promising practical applications. Within synchronization analysis, the synchronization bandwidth serves as a pivotal metric. While current research predominantly focuses on symmetric evaluations of synchronization bandwidth, the investigation into potential asymmetries within nonlinear oscillators remains unexplored, carrying implications for sensor application performance. This paper conducts a comprehensive exploration employing straight and arch beams capable of demonstrating linear, hardening, and softening characteristics to thoroughly scrutinize potential asymmetry within the synchronization region. Through the introduction of weak harmonic forces to induce synchronization within the oscillator, we observe distinct asymmetry within its synchronization range. Additionally, we present a robust theoretical model capable of fully capturing the linear, hardening, and softening traits of resonators synchronized to external perturbation. Further investigation into the effects of feedback strength and phase delay on synchronization region asymmetry, conducted through analytical and experimental approaches, reveals a consistent alignment between theoretical predictions and experimental outcomes. These findings hold promise in providing crucial technical insights to enhance resonator performance and broaden the application landscape of MEMS (Micro-Electro-Mechanical Systems) technology.

## 1. Introduction

Synchronization is a pervasive phenomenon in nature, observed across a wide spectrum of systems, including quantum systems [1,2], nano/microstructures [3,4,5,6,7], and artificial or biological clusters [8,9,10,11]. Investigations on synchronization trace back to Huygens’ work in 1677 [12]. Presently, synchronization exhibits remarkable performance in sensing, attributed to exceptional frequency stability [13,14,15,16] and the ease of frequency recognition [17]. This has found significant application in microelectromechanical systems (MEMS) owing to their compact size and compatibility with integrated circuit fabrication [18,19,20,21]. However, it is crucial to recognize that, due to size effects and intrinsic electrostatic excitation mechanisms, MEMS systems often operate within nonlinear regimes [22], which can exhibit a broader range of dynamic phenomena compared to the linear regime.

Nonlinearities wield a substantial influence on synchronization dynamics. For instance, they impact the synchronization process by shaping energy potential wells [23], allowing for accelerated synchronization processes and compressed synchronization time through phase delay and feedback strength modulation. Nonlinear effects can also affect resonator frequency stability via nonlinear amplitude-frequency dependences [24]. By manipulating the oscillator’s operating point, achieving a zero-dispersion state enhances frequency stability significantly. Most notably, nonlinearities can substantially broaden synchronization bandwidth compared to linear oscillators by orders of magnitude [25,26,27]. Manipulating feedback, phase delay, coupling, and perturbation strengths significantly affects synchronization behavior.

While ample evidence suggests nonlinearity’s impact on synchronization behavior, existing studies mostly assume synchronization regions to be symmetric to the self-oscillation frequency [28,29,30]. Yet, due to the amplitude-frequency effect [24] induced by nonlinearity, synchronization regions can exhibit asymmetry [31]. Such results have been shown in the experimental measured synchronization region in [26,27]; however, the asymmetry of synchronization has been largely unexplored regarding the intrinsic mechanism of nonlinearity’s influence. Asymmetric synchronization in systems such as communication networks or electronic circuits can result in decreased efficiency and predictability during data transmission or signal processing, and it may exert notable effects on the stability and overall performance of engineering [32] and physical systems [33]. Investigating the symmetry of nonlinear oscillators not only aids in understanding intrinsic nonlinear synchronization mechanisms but also enables predictive control, optimizing sensor capabilities [34].

This paper introduces self-oscillation oscillators designed to demonstrate versatile behaviors—linear, hardening, and softening—to synchronize with external perturbation signals. Our investigation reveals an asymmetric synchronization region resulting from induced nonlinearity. Furthermore, we present a comprehensive theoretical model that elucidates the intrinsic mechanisms governing the impact of nonlinearity on the synchronization region.

The following sections of this paper are organized as follows: Section 2 outlines the experimental setup and signal processing, Section 3 presents the key findings from three distinct oscillators, Section 4 explores theoretical modeling and analytical discoveries, and Section 5 provides conclusive remarks.

## 2. Materials and Methods

### 2.1. Device Characteristics

The nonlinearity, delineated as hardening and softening based on the amplitude-frequency response, is pivotal in our investigation of synchronization region symmetry. To comprehensively explore this effect, we employed two distinct resonators: the clamped-to-clamped (C-C) straight beam R_s_ (Figure 1a) and the C-C arch beam R_a_ (Figure 1b), which were manufactured from silicon by the commercial company MEMSCAP. These resonators exhibit hardening and softening behaviors under strong excitation strength, respectively. However, under weak excitation strength, both demonstrated linear characteristics. Each resonator features two side electrodes for excitation and detection, with all anchor electrodes grounded. The dimensions of the resonators are shown in Table 1.

### 2.2. Excitation and Signal Processing Setup

For excitation, we applied a combination of DC bias voltage ($V_{\mathrm{DC}}$, fixed at 20 V in all experiments) and an AC voltage ($V_{\mathrm{AC}}$) to the excitation electrode, while the beam body is grounded. The resultant electrostatic force, generated by the dynamic potential difference between the excitation electrode and the beam, induces corresponding vibration in the resonator (Figure 1c). In the detection process, the potential difference between the beam and the detection electrode was maintained at $V_{\mathrm{DC}}$. As the beam vibrates, the capacitance (*C*) of the comb finger changes (shown in the inset of Figure 1c), leading to a variation of the total charge Δ*Q* = *V_DC_* · Δ*C*. This variation generates current that characterizes the vibration of the beam. This vibration signal was processed through a differential amplifier (Texas Instruments OPA 656U), serving dual purposes: converting the current to voltage and amplifying the signal, and mitigating inherent feedthrough impact by combining with the equivalent capacitance [35] and the local oscillator 1. Subsequently, the processed signal is fed into a lock-in amplifier (Zurich Instruments, HF2LI), equipped with a built-in phase feedback circuit capable of real-time tuning of phase delay ($\phi_{0}$) and feedback strength ($f_{0}$) to construct the self-oscillating oscillator. The open-loop and closed-loop responses can be obtained by turning off and on the built-in phase-locked loop (PLL) circuit. All experiments are conducted within a vacuum chamber, maintaining a pressure below 0.1 Pa to minimize air damping losses, as shown in Figure 1c.

### 2.3. Establishment and Measurement of Synchronization

Our focus is on exploring synchronization region symmetry in resonators exhibiting both hardening and softening nonlinearities. We initially characterized the driving voltage ranges for different states of the beams, including linear, hardening, and softening, based on the open-loop amplitude-frequency responses. Subsequently, by utilizing the pre-established phase-locked loop (PLL) within the lock-in amplifier, we were able to induce self-oscillation to the resonator. Furthermore, we assumed that once the oscillation was established, the oscillator would remain in the same state as the open-loop response. Specifically, we delve into injection synchronization, wherein an oscillator synchronizes with an external perturbation force. In our experiments, we initially established the self-oscillating oscillator by activating the PLL circuit. Subsequently, we inject a perturbation signal from the function generator (purple region) into the oscillator. The perturbation signal is transferred into the internal closed-loop control circuit of the lock-in amplifier. It is then combined with the feedback force generated by local oscillator 2 and simultaneously applied to the beam. As the perturbation signal frequency aligns within the synchronization region around the oscillator’s self-oscillation frequency, synchronization is established, causing the oscillator’s frequency to align with the external perturbation. By systematically varying the perturbation frequency around the self-oscillation frequency, we delineated the synchronization region.

## 3. Results

### 3.1. Synchronization Region of the Linear Oscillator

Our analysis initially focused on the symmetry within the synchronization region of a linear oscillator, utilizing a straight C-C beam displaying linear characteristics under minimal driving strength, depicted in Figure 2a. To comprehensively explore this region, we conducted perturbation frequency sweeps backward and forward around the oscillation frequency. During the backward sweep, the oscillation frequency promptly synchronizes with the perturbation, subsequently decreasing until desynchronization occurs at $\Omega_{1}$. Conversely, during the forward sweep, desynchronization takes place at $\Omega_{2}$. Consequently, the upper bound ($\Omega_{2}$) and lower bound ($\Omega_{1}$) of the synchronization region are quantified, as illustrated in Figure 2b. We additionally represented the measured frequency ratio and phase difference between the perturbation and the oscillator in Figure 2c,d. These figures elucidate the occurrence of frequency and phase locking phenomena during synchronization.

Defining the frequency difference between the synchronization lower bound $\Omega_{1}$ and the center frequency $\Omega_{0}$ as the lower half bandwidth (Δ*Ω*_1_ = *Ω*_0_ − *Ω*_1_), and the frequency difference between the synchronization upper bound *Ω*_2_ and the center frequency *Ω*_0_ as the upper half bandwidth (Δ*Ω*_2_ = *Ω*_2_ − *Ω*_0_), we plotted the upper and lower bounds of the synchronization region, varied with the phase delay and observed in Figure 2e, which revealed noteworthy trends. Notably, a monotonic decrease in synchronization bandwidth with phase delay is evident, particularly when employing a small feedback strength ($V_{f}=10\;\mathrm{mV}$), reaching its minimum at $\phi_{0}=\pi /2$, as previously verified in [26,27]. Importantly, throughout the entire range of phase delay variations, the lower half bandwidth Δ*Ω*_1_ and upper half bandwidth Δ*Ω*_2_ consistently maintained equal magnitudes (Figure 2f), signifying the symmetry of the synchronization region under linear conditions.

### 3.2. Synchronization Region of the Hardening Oscillator

As the excitation gradually intensified, *R_s_* displayed hardening nonlinearity, showcasing an increase in the response frequency with the amplitude, depicted in Figure 3a. The measured upper and lower bounds of the synchronization region, illustrated in Figure 3b,c, bring to light an intriguing observation from Figure 3b: the considerable discrepancy between the upper half bandwidth of 47.3 Hz and the lower half bandwidth of 32.5 Hz, indicating the asymmetry within the synchronization region under hardening nonlinear conditions.

In Figure 3c, the curves demonstrate the synchronization region’s progression concerning the phase delay. Notably, a continuous augmentation of the synchronization region is observed with increasing phase delay, culminating at its maximum value at a phase delay of 90°.

Moreover, Figure 3d charts the variation in the difference between the upper half bandwidth and the lower half bandwidth ($\Delta \Omega_{1}-\Delta \Omega_{2}$) as a function of phase delay. Clearly, $\Delta \Omega_{1}-\Delta \Omega_{2}$ consistently maintains a positive value within the phase delay range from 30° to 90°.

### 3.3. Synchronization Region of the Softening Oscillator

Within the domain of softening oscillators, we employed an arch beam known for its pronounced softening nonlinearity, particularly evident under excitations ranging from 100 to 300 mV, as depicted in Figure 4a. In this range, we measured the upper and lower limits of the synchronization region, unveiling an intriguing observation: an asymmetric synchronization region. However, unlike the scenario in hardening nonlinear cases, here, the upper half bandwidth of 84.1 Hz is notably smaller than the lower half bandwidth of 108.8 Hz.

Figure 4c illustrates the synchronization region’s behavior across varying phase delays. Notably, the lower half bandwidth consistently maintains a larger value than the upper half bandwidth throughout the phase delay range from 30° to 90°. Moreover, Figure 4d delineates the relationship between the difference of the upper half bandwidth and the lower half bandwidth ($\Delta \Omega_{1}-\Delta \Omega_{2}$) as a function of feedback strength. Remarkably, as the feedback strength escalates from 100 mV to 250 mV, this asymmetrical phenomenon becomes increasingly pronounced.

## 4. Theoretical Modeling

In pursuit of a comprehensive understanding of the inherent mechanisms governing the symmetry within the synchronization region of nonlinear oscillators, we employ the following governing equation. This equation adequately captures and characterizes the linear, hardening, and softening traits observed in the aforementioned resonators:(1)$$\begin{array}{ll} \mathrm{EI}\frac{\partial^{4}\hat{w}}{\partial {\hat{x}}^{4}} & +(\rho S+\delta (\hat{x}-\frac{L}{2}))\frac{\partial^{2}\hat{w}}{\partial {\hat{t}}^{2}}+\hat{c}\frac{\partial \hat{w}}{\partial \hat{t}}\\ & =[\frac{\partial^{2}\hat{w}}{\partial {\hat{x}}^{2}}+\frac{d^{2}{\hat{w}}_{0}}{d{\hat{x}}^{2}}][\frac{\mathrm{ES}}{2L}\int_{0}^{L}\{{\left(\frac{\partial \hat{w}}{\partial \hat{x}}\right)}^{2}+2(\frac{\partial \hat{w}}{\partial \hat{x}}\frac{d{\hat{w}}_{0}}{d\hat{x}})\}d\hat{x}]-\hat{F}(\hat{w},\hat{\Omega }\hat{t})\delta (\hat{x}-\frac{L}{2}), \end{array}$$
where E, $I=bh^{3}/12$, ρ, S=bh, $\hat{c}$ are the effective Young’s modulus, moment of inertia, mass density, cross section area, and damping coefficient of the beam, respectively. δ is the Dirac delta function, and $\hat{F}\left(\hat{w},\hat{\Omega }\hat{t}\right)$ is the electrostatic force loaded on the beam body, which is generated by the comb tooth electrode [27],
(2)$$\begin{array}{c} \hat{F}\left(\hat{w},\hat{\Omega }\hat{t}\right)=\frac{1}{2}\frac{\partial C_{1}}{\partial \hat{w}}{\left(V_{AC}\sin\left(\hat{\Omega }\hat{t}\right)+V_{DC}\right)}^{2}+\frac{1}{2}\frac{\partial C_{2}}{\partial \hat{w}}V_{DC}^{2}, \end{array}$$
where
$$\begin{array}{c} C_{1}=\frac{2Nah\varepsilon_{0}}{d-\hat{w}}+\frac{2N\left(c+\hat{w}\right)h\varepsilon_{0}}{g};\\ C_{2}=\frac{2Nah\varepsilon_{0}}{d+\hat{w}}+\frac{2N\left(c-\hat{w}\right)h\varepsilon_{0}}{g}. \end{array}$$
*N* is the number of the fingers, $\varepsilon_{0}$ is the dielectric constant, and h, a, c, d, g are the thickness, width of the finger, the initial comb finger overlap, the initial spacing between the combs and the proof mass, and the gap spacing between the fingers, respectively. In this work, we utilize the Galerkin method to develop a reduced-order model. By incorporating self-oscillation and adding perturbation injection, we are able to derive the resulting governing equation [27],
(3)$$\ddot{x}+Q^{-1}\dot{x}+\omega_{0}^{2}x+\alpha x^{2}+\beta x^{3}=f_{0}\cos\left(\phi +\phi_{0}\right)+f_{s}\cos\left(\Omega_{s}t\right),$$
where Q is the quality factor of the oscillator, measuring the ratio between the decay time due to damping and the oscillation period. α is the normalized quadratic nonlinearity coefficient, β is the normalized cubic nonlinearity coefficient, $f_{0}$ is the normalized feedback strength, $f_{s}$ is the normalized perturbation strength, and $\Omega_{s}\;$represents the normalized perturbation frequency. Let $K=\frac{3}{8}\beta^{3}-\frac{5}{12}\alpha^{2}$ be the equivalent nonlinearity, wherein K=0 signifies linearity, K>0 indicates hardening nonlinearity, and K<0 represents softening nonlinearity [36].

Initially, we analyze the self-sustained oscillator, characterized by $f_{s}=0$. Utilizing the expression $x\left(t\right)=A\;\cos\phi =A\;\cos\left(\Omega t+\theta \right)$, where A represents the instantaneous amplitude and θ the initial phase, and by applying the method of multiple scales, we determine the amplitude ($A_{0}$) and frequency ($\Omega_{0}$) of the self-sustained oscillation.
(4a)$$A_{0}=\frac{Qf_{0}\sin\left(\phi_{0}\right)}{\Omega_{0}};$$
(4b)$$\Omega_{0}=\sqrt{\omega_{0}^{2}+\frac{4Q^{2}K^{2}}{A_{0}^{4}}}.$$

When an external perturbation is introduced to the self-sustained oscillator, it perturbs the oscillator around its self-oscillation state. Letting $\varphi_{2}$ represent the phase difference between the oscillator and external perturbation. Equation (3) can be solved using the perturbation method, revealing that the deviation of the phase difference exhibits correlations with nonlinear terms. Letting $\Omega =\omega_{0}+\epsilon \delta_{1}$, $\Omega_{s}=\omega_{0}+\epsilon \delta_{2}$, we have the deviation of the phase difference,
(5)$$\varphi_{2}^{'}=\delta_{3}-\delta_{2}+K+N_{1}\sin\left(\varphi_{2}\right)+N_{2}\cos\left(\varphi_{2}\right),$$
where
$$N_{1}=\frac{f_{0}\cos\left(\phi_{0}\right)}{2A_{0}\Omega_{0}}+\frac{f_{s}\sin\left(\phi_{0}\right)}{2Qf_{0}}+\frac{f_{s}K\cos\left(\phi_{0}\right)}{f_{0}};$$
$$N_{2}=-\frac{f_{0}\sin\left(\phi_{0}\right)}{2A_{0}\Omega_{0}}-\frac{f_{s}\cos\left(\phi_{0}\right)}{2Qf_{0}}+\frac{f_{s}K\sin\left(\phi_{0}\right)}{f_{0}}.$$

$\varphi_{2}$ is the phase difference between the oscillator and external perturbation. When synchronization is established, $\Omega_{s}=\Omega$; thus, $\delta_{1}=\delta_{2}$. The simplified function is a typical Adler equation; the lower and upper bounds are as follows:(6a)$$\Omega_{1}=\Omega_{0}+K-\sqrt{N_{1}^{2}+N_{2}^{2}};$$
(6b)$$\Omega_{2}=\Omega_{0}+K+\sqrt{N_{1}^{2}+N_{2}^{2}}.$$

The derived results illustrate a clear relationship between the upper and lower bounds of the interval and the equivalent nonlinearity, resulting in asymmetry within the interval. Figure 5 showcases the normalized theoretical simulations of the upper and lower bounds of the synchronization region concerning the equivalent nonlinearity, $f_{0}\;$ magnitude, and phase delay $\phi_{0}$, aligning remarkably well with experimental observations. The parameters used in these simulations were derived from the experimental data and subsequently normalized. Consistent with prior research, the synchronization region expands with increasing driving voltage.

Under linear conditions, the synchronization region maintains symmetry, with its bandwidth decreasing alongside phase delay. Conversely, in oscillators with higher equivalent nonlinearity (i.e., exhibiting hardening nonlinearity), the frequency difference $\Delta \Omega_{2}$ from the upper bound $\Omega_{2}$ to the center frequency $\Omega_{0}$ surpasses $\Delta \Omega_{1}$ from the lower bound $\Omega_{1}$ to $\Omega_{0}$. Increasing feedback force $f_{0}$ gradually raises the center frequency, leading to a nonlinear increase in synchronization bandwidth, peaking at a phase delay of π/2.

For oscillators with lower equivalent nonlinearity (softening nonlinearity), amplifying the feedback force $f_{0}$ gradually lowers the center frequency [37]. Notably, $\Delta \Omega_{1}$ from the lower bound $\Omega_{1}$ to $\Omega_{0}$ significantly exceeds $\Delta \Omega_{2}$ from the upper bound $\Omega_{2}$ to $\Omega_{0}$. This asymmetry intensifies with higher feedback strength and prolonged phase delay.

## 5. Conclusions

This study delves into a comprehensive exploration, both theoretically and experimentally, unraveling the intricacies of asymmetry within the synchronization region in resonators featuring hardening and softening equivalent nonlinearities. Our investigation has culminated in the derivation of an equivalent nonlinearity form, illuminating distinct symmetries inherent in hardening and softening phenomena. Notably, we observed that in the case of hardening, the upper bound prevails, whereas in softening scenarios, the lower bound dominates, signifying differing symmetry profiles.

Moreover, our scrutiny extended to examining the influence exerted by the magnitude of the driving force and the phase delay on asymmetry. Our findings elucidate that as these variables increase, the asymmetry within the synchronization region becomes more pronounced, highlighting their critical role in shaping synchronization dynamics.

Our experimental results substantiate the theoretical predictions, particularly evident under small external excitation forces, reinforcing the robustness of our theoretical framework. The observed asymmetry within the synchronization region bears significant implications for system performance and accuracy. Understanding these asymmetries holds paramount importance in devising, controlling, and optimizing synchronization systems.

Looking ahead, the potential lies in harnessing and manipulating this asymmetry to further elevate resonator performance, thereby expanding the application spectrum of MEMS technology. By gaining mastery over this asymmetry, future advancements could potentially unlock new frontiers, enhancing the precision and versatility of MEMS-based devices.

## Figures and Tables

**Figure 1 micromachines-15-00238-f001:**
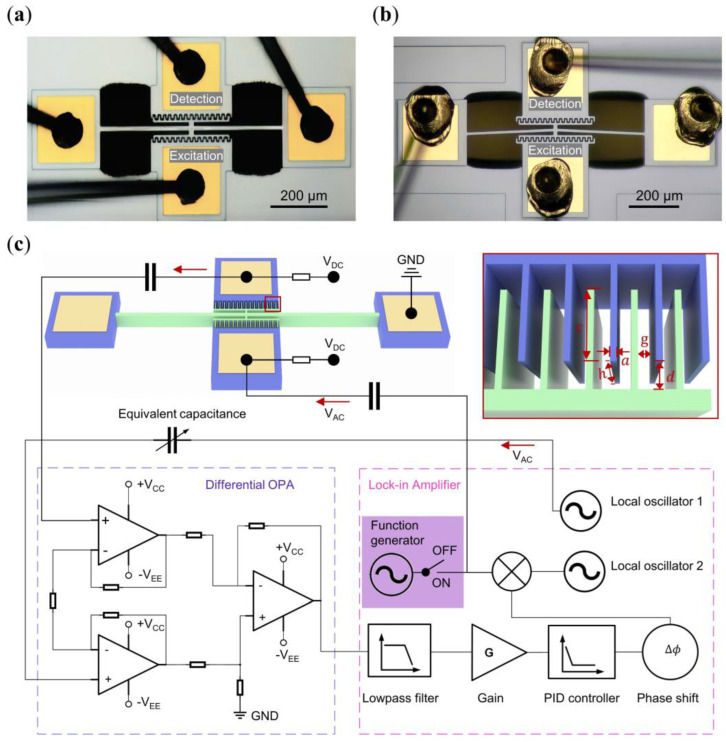
Device characteristics and experimental setup. (**a**) Straight beam (linear and hardening nonlinearity); (**b**) arch beam (softening nonlinearity); (**c**) experimental setup of the phase feedback loop. Inset depicts the comb tooth electrode.

**Figure 2 micromachines-15-00238-f002:**
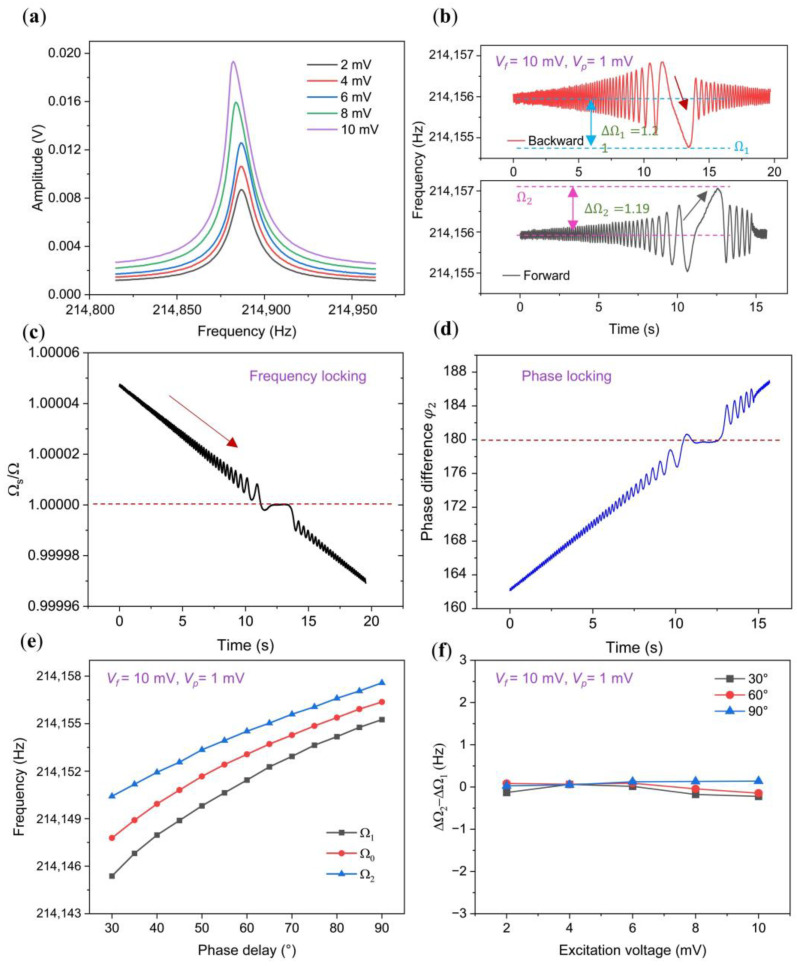
Characteristics of the synchronization region in a linear oscillator. (**a**) Open-loop amplitude-frequency response for a linear straight beam with increasing driving force from 2 mV to 10 mV; (**b**) response under backward and forward sweeping at *V_f_* = 10 mV. The double-headed arrow shown in this figure indicates the range, while the single-headed arrow represents the sweep direction of the perturbation frequency; (**c**) ratio between the perturbation frequency and the oscillator’s frequency measured from the backward sweep frequency results in (**b**); (**d**) phase difference between the perturbation and the oscillator measured from the backward sweep frequency results in (**b**); (**e**) synchronization region observed at *V_f_* = 10 mV and *V_p_* = 1 mV, with the upper limit, central frequency, and lower limit depicted by blue, red, and black lines, respectively; (**f**) analysis of the differences between upper and lower half bandwidths.

**Figure 3 micromachines-15-00238-f003:**
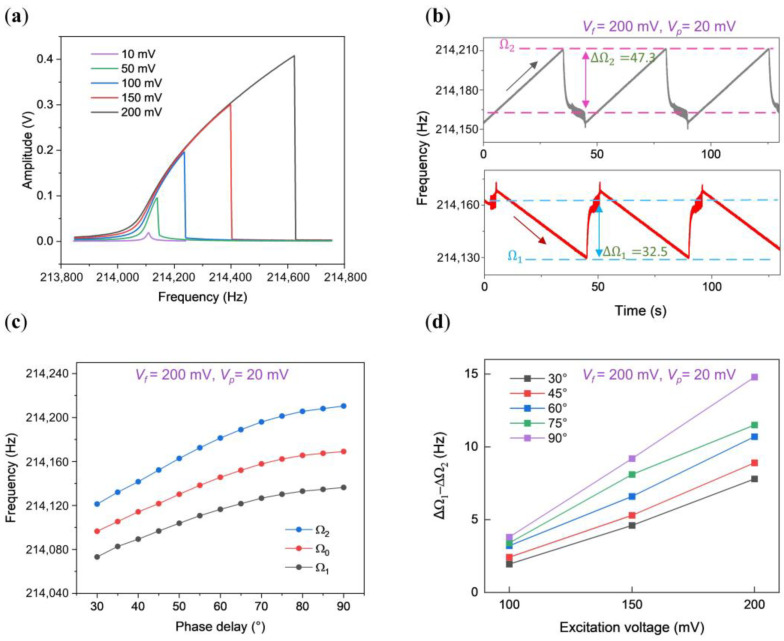
Characteristics of the synchronization region in a hardening nonlinear oscillator. (**a**) Open-loop amplitude-frequency responses of the straight beam with driving force increasing from 10 mV to 200 mV; (**b**) response under forward and backward sweeping at *V_f_* = 200 mV and *V_p_* = 20 mV. The double-headed arrow shown in this figure indicates the range, while the single-headed arrow represents the sweep direction of the perturbation frequency; (**c**) observed synchronization region at various phase delays, with *V_f_* = 200 mV and *V_p_* = 20 mV, showcasing the upper limit, central frequency, and lower limit as blue, red, and black lines, respectively; (**d**) variation in the half bandwidths’ difference at phase delays of 30°, 60°, and 90°.

**Figure 4 micromachines-15-00238-f004:**
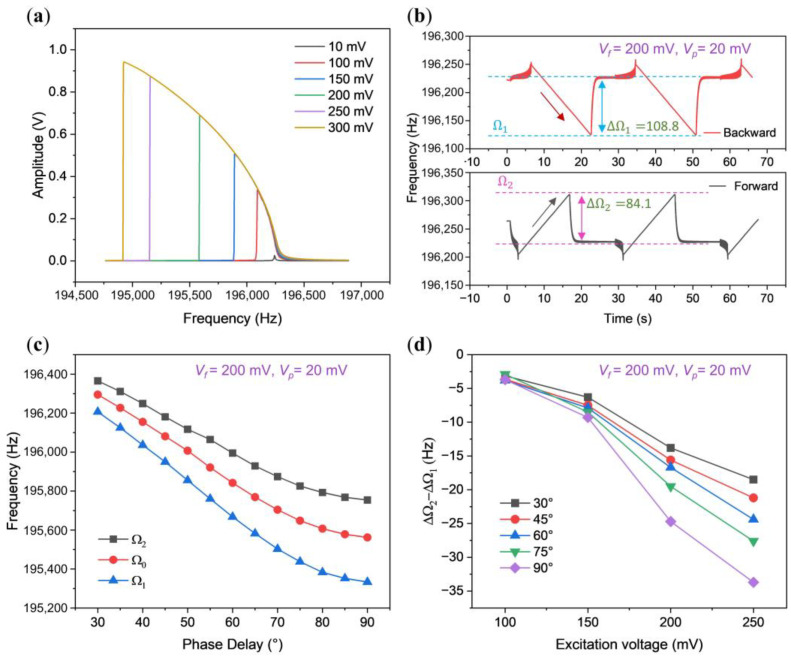
Characteristics of the synchronization region in a softening nonlinear oscillator. (**a**) Open-loop amplitude-frequency responses of the arch beam with increasing driving force from 10 mV to 300 mV; (**b**) response under both backward and forward sweeping at *V_f_* = 200 mV and *V_p_* = 20 mV. The double-headed arrow shown in this figure indicates the range, while the single-headed arrow represents the sweep direction of the perturbation frequency; (**c**) synchronization region at *V_f_* = 200 mV and *V_p_* = 20 mV, varying with phase delay, represented by the upper limit, central frequency, and lower limit as blue, red, and black lines, respectively; (**d**) analysis of the difference between upper and lower half bandwidths.

**Figure 5 micromachines-15-00238-f005:**
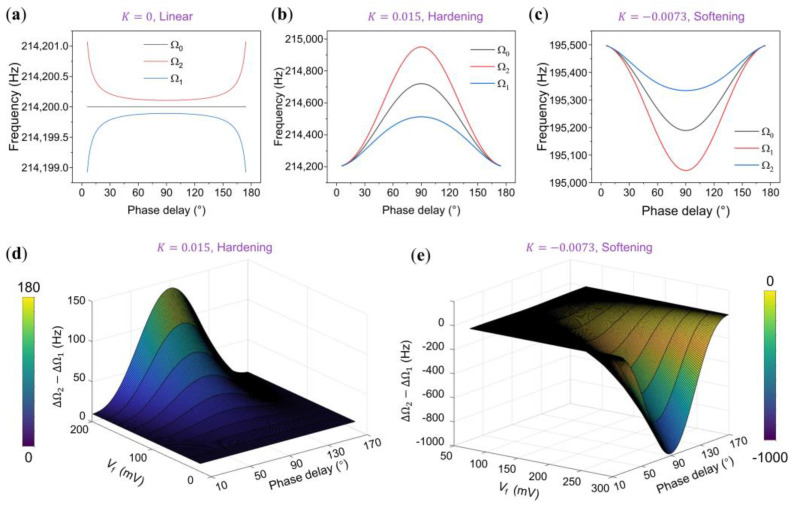
Simulations of the synchronization characteristics. (**a**–**c**) depict the upper limit, self-oscillation frequency, and lower limit of the synchronization region, respectively, each varied with the phase delay $\phi_{0}$. In (**a**), the oscillator operates in a linear state, with parameters normalized from the 10 mV open-loop amplitude response shown in Figure 2a. Here, *Q* = 10,000,$\;f_{0}$ = 1.3 × 10^−6^, $f_{s}$ = 1.3 × 10^−7^, and *K* = 0. (**b**) shows the oscillator in a hardening state, with parameters derived from the 300 mV open-loop amplitude response in Figure 3a, characterized by *Q* = 10,000, $f_{0}$ = 4 × 10^−5^, $f_{s}$ = 4 × 10^−6^, and *K* = 0.015. In (**c**), the oscillator functions in a softening state, with parameters normalized from the 10 mV open-loop amplitude response in Figure 4a, where *Q* = 10,000, $f_{0}$ = 1 × 10^−4^, $f_{s}\;$= 1 × 10^−5^, and *K* = −0.0073. (**d**,**e**) present numerical simulations of $\Delta \Omega_{1}-\Delta \Omega_{2}$ in straight and arch beams, respectively, as they respond to phase delay changes with the feedback force $V_{f}$ varying from 20 to 200 mV and 50 to 300 mV. In these cases, the perturbation $V_{p}$ is set to 0.1$V_{f}$. The other parameters are consistent with those in (**b**,**c**).

**Table 1 micromachines-15-00238-t001:** Parameters of the MEMS resonator.

Beam	Length	Width	Thickness	Arch Height
straight beam R_s_	482 μm	12 μm	25 μm	
arch beam R_a_	640 μm	5 μm	25 μm	7 μm

## Data Availability

Data are contained within the article.

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
