# Peer review of "Nonlinearity-Induced Asymmetric Synchronization Region in Micromechanical Oscillators"

_micromachines, 2024, doi:10.3390/mi15020238_

Round 1
Reviewer 1 Report
Comments and Suggestions for Authors
1 Figure 1 gives a microscopic picture of the resonator, which is unfriendly to readers who are not familiar with this field, and the author should give a diagram of the working principle of the resonator, introducing how the resonator works, especially how the perturbation frequency is applied.
2 What is the voltage in the legend in Figure 2a? Is it VDC? What is the value of VAC, and what do Vf and Vp represent?
3 Note that the unit of amplitude in Figure 2a is V, what is the relationship between the voltage and the actual amplitude?
4 How is the frequency in Figure 2b measured? Normally, the waveform of the voltage is measured, and the frequency can be obtained by Fourier transform, but here the frequency is time-changing, how can you get the accurate frequency at each moment?
5 What are the values of the parameters in Eq. 1 and how do they relate to the actual physical parameters?
6 Figure 5 shows the simulation results, do the simulation results match the experimental results?
7 Note the ordinate of Figure 5 is the frequency, the size does not exceed 1 Hz, and the experimental results are greater than 100,000 Hz. Why is there such a big difference?
8 what are the definitions of 𝛺0, 𝜙0, 𝜙2, Vf, Vp, upper and lower bounds et al?
09 In Fig.2(a), different curves can be identified by legend. However, it is not clear that which value in the legend is corresponding to other three subplots?
Comments on the Quality of English Language
Genereall speaking, passive voice and simple past tense are preferred in scientific writing
The sentence in the second paragraph in Page 4 is not completed.
The introduction to related works is too simple
Reviewer 2 Report
Comments and Suggestions for Authors
The current version of the manuscript cannot be accepted unless authors can address following questions:
1. On page 4, Fig. 2b, please explain how to measure this frequency versus time. It is quite confusing. What is the frequency shown in Fig.2b ? is it the perturbation frequency or the resonator oscillation frequency ? Also, what is the resonant state of the resonator in the synchronous region ? is it still in the linear response state or in the self-oscillating state ? if the latter, please specify the method to drive it into the oscillating state. Many important details are missing.
2. The key point of synchronization phenomenon in MEMS is the lock of frequency and phase. Therefore, please analyze the frequency locking between the perturbation frequency and also the phase difference between the external perturbation frequency and oscillation frequency, \phi_2. It is important, but it is missing in this manuscript.
3. Besides, it is also important to address the amplitude of the external perturbation signal. It also plays a key role in inducing the synchronization phenomenon.
Reviewer 3 Report
Comments and Suggestions for Authors
The authors present experimental and theoretical study on the asymmetry behavior of non-linear mechanical systems. It will be of interest to the field of MEMS. I have some questions as below:
1. i couldn't understand how the perturbation frequency is swept backward and forward around the oscillation frequency. From the figure 2, they both oscillate back and forth around the center frequency. What is the difference between the two cases of backward and forward?
2. For sentence in line 196-198, i can not find correspondence of sustaining force and external perturbation in this sentence.
3, what is the expression for w0?
4. what is the definition of epsilon and delta in line 221?
5. "the synchronization region maintains symmetry, with its 233 bandwidth decreasing alongside phase delay", however from figure 5a, the distance between upper and lower limit increases as the phase increase. which is opposite to the experiment. Do you have any idea?
Reviewer 4 Report
Comments and Suggestions for Authors
The paper is about nonlinearity-induced asymmetric synchronization region in MEMS devices.
Authors presented and compared theoretical study as well mechanical characterization of two different MEMS structures – beams - straight and arched.
Paper is well prepared, all sections have adequatecontent, are logical etc…
References, introduction as well conclusions, figures are presented in suitable form.
Few comments, that Authors can comment/explain/update in the paper:
- Beam geometry, arched one is longer – it is clear, but why there is a different width?
- MEMS device material – silicon? or different?
There is several software tools (Comsol, Coventorware, MEMS+, Ansys) – will be nice to have some comparison with modeling and simulation results – as the structures are quite simple…just comment rather for next paper.
I’m not sure about significance of paper and difficult to judge “Interest to the readers” as I saw a very big amount of similar topics papers in the past.
Round 2
Reviewer 1 Report
Comments and Suggestions for Authors
All the comments are well responsed. I think it can be accepted.
Author Response
All the comments are well responsed. I think it can be accepted.
Response: Thanks for the referee. We have thoroughly reviewed the manuscript and implemented several modifications to ensure that all methods and results are comprehensively described. Additionally, we have verified that the conclusions are supported by the presented results.
Thank you for your positive feedback and for considering our work for acceptance. We appreciate the time and effort you took to review our responses to the comments. Please let us know if there are any further steps or information required from our side. Thank you once again for your support and guidance throughout this process.